# Recruitment of the default mode network during a demanding act of executive control

**Ben M Crittenden[1,2,3]\*, Daniel J Mitchell[1], John Duncan[1,2]**

[1]Medical Research Council Cognition and Brain Sciences Unit, Cambridge, United Kingdom; [2]University of Cambridge, Cambridge, United Kingdom; [3]Oxford Centre for Human Brain Activity, University of Oxford, Oxford, United Kingdom

**Abstract** In the human brain, a default mode or task-negative network shows reduced activity during many cognitive tasks and is often associated with internally-directed processes, such as mind wandering and thoughts about the self. In contrast to this task-negative pattern, we show increased activity during a large and demanding switch in task set. Furthermore, we employ multivoxel pattern analysis and find that regions of interest within default mode network are encoding task-relevant information during task performance. Activity in this network may be driven by major revisions of cognitive context, whether internally or externally focused.

## Introduction

Functional magnetic resonance imaging (fMRI) has repeatedly demonstrated that cognitive tasks of many kinds decrease activity in a large-scale cortical network, variously termed the task-negative or default mode network (DMN) (*Shulman et al., 1997*; *Raichle et al., 2001*; *Andrews-Hanna et al., 2010*). The DMN consistently shows reduced activity during task performance compared to rest (*Raichle and Snyder, 2007*) and often reduced activity for harder compared to easier task versions (*Gilbert et al., 2012*). In contrast to this general pattern, increased activity has been reported in a cluster of mental states involving thinking about the self, one's own perspective, or the perspective of others (*Buckner and Carroll, 2007*). Examples include recollecting previous experiences (*Vincent et al., 2006*) or imagining future ones (*Addis et al., 2007*), mind-wandering (*Mason et al., 2007*), and theory of mind tasks (*Young et al., 2010*). The DMN has thus been linked to a number of high-level cognitive processes, such as self-referential processing (*Gusnard et al., 2001*) and imaginary scene construction (*Hassabis and Maguire, 2007*).

Recently, *Andrews-Hanna et al. (2010)* have argued that the DMN separates into three sub-networks. Using graph theoretical analytic approaches to resting-state fRMI data, Andrews-Hanna et al. identified a core sub-network comprising bilateral posterior cingulate cortex (PCC) and anterior medial prefrontal cortex (AMPFC) a medial temporal lobe (MTL) sub-network made up of ventromedial prefrontal cortex (VMPFC), bilateral hippocampal formation (HF), parahippocampus (PHC), retrosplenial cortex (Rsp), and posterior inferior parietal lobule (pIPL) and a dorsomedial prefrontal cortex (DMPFC) sub-network which includes the DMPFC, bilateral temporal parietal junction (TPJ), lateral temporal cortex (LTC), and the temporal Pole (TempP). Andrews-Hanna et al. argue for a degree of functional segregation between these sub-networks, with the MTL sub-network especially linked to construction of mental scenes based on memory, and the DMPFC network more involved in mentalising. These segregations, however, are likely to be relative, with many experiments, for example, linking all three DMN sub-networks to conscious recollection (*Vilberg and Rugg, 2012*).

Here, we consider a simple conceptualisation of DMN function and apply it within the Andrews-Hanna framework. Arguably, imagination, mind-wandering, and taking another's perspective have in

\*For correspondence: ben.crittenden@psych.ox.ac.uk

**Competing interests:** The authors declare that no competing interests exist.

**Reviewing editor**: David C Van Essen, Washington University in St Louis, United States

**eLife digest** The default mode network is a network in the brain that is often active when we think about ourselves, reminiscence about the past or just let our minds wander. However, this network—which involves many different regions of the brain—usually becomes inactive when we focus on a specific cognitive task.

Now Crittenden et al. have used a technique called functional MRI to show that the default mode network can become active again if we switch from one task to another. Functional MRI works by measuring the blood flow in the brain: regions of the brain that are active have more blood flow than regions that are not active.

Crittenden et al. studied the brains of human subjects as they performed a series of different tasks. These experiments showed that the activity of the default mode network does not change when the subject is focused on a single task. This is also true for when the subject switches between two similar tasks. However, when the subject switches between two very different tasks, the network becomes significantly more active. Moreover, the patterns of activity in the network seem to reflect the nature of the tasks.

The work of Crittenden et al. strongly suggests that in order to successfully switch between two different tasks, the brain needs to engage the default mode network and allow the mind to wander. Future studies will involve exploring how different the two tasks need to be in order to activate the default mode network, and studying how brain damage within the network may impair patients ability to switch between different tasks.

common a substantial change from the current cognitive context. Similarly, conscious recollection is typically conceived as reactivation of a previously-experienced episode, with components linked into a complex surrounding, context. Substantial shifts of context may be common in everyday activity, for example, a shift from cooking dinner to giving directions to guests over the phone, but less common in the constrained setting of typical laboratory tasks. For example, in a recent review of neuroimaging studies of task switching (*Kim et al., 2011*), tasks that they argue required a contextual switch involved either a change in simple attended features or binary categorization rules using a fixed, small set of possible stimuli. Irrespective of specific high-level processes involved, we reasoned that the DMN may be involved in any large switch of cognitive context—an operation presumably calling for relaxation of many aspects of a current attentional focus, with concomitant activation of representations and processes relevant to the new context.

To test this hypothesis, we used a novel experimental paradigm that required participants to switch between similar and dissimilar tasks, within a relatively large set of six tasks (*Figure 1*). The six tasks were each associated with a different rule, as determined by the colour border surrounding the task stimulus. The tasks were split into three groups defined by stimulus category, with two possible tasks per stimulus type. A no-switch trial occurred when participants had to apply the same rule that was applied on the previous trial. A similar-task-switch trial—resembling switches in typical neuroimaging studies—occurred when participants had to apply the other rule from the same category as the previous trial. A dissimilar-task-switch occurred when participants had to apply a rule from a different category compared to the previous trial.

Contrary to the common concept of a task-negative system, we predicted increased DMN recruitment for the most difficult condition of switching between dissimilar tasks. We found this to be the case and furthermore, that the activity increase was selectively found in the Core and MTL sub-networks. In addition, we provide evidence that all sub-networks of the DMN represented task-related information during task performance.

## Results

### Behavioral results

Accuracy on all tasks was high (median accuracy for all tasks >95%, inter quartile range <6%). As predicted, response time (RT) was significantly longer when switching between dissimilar tasks (2043 ms) compared both to trials when no task switch occurred (1670 ms; $t_{17} = 8.6$, p < 0.001), and to trials

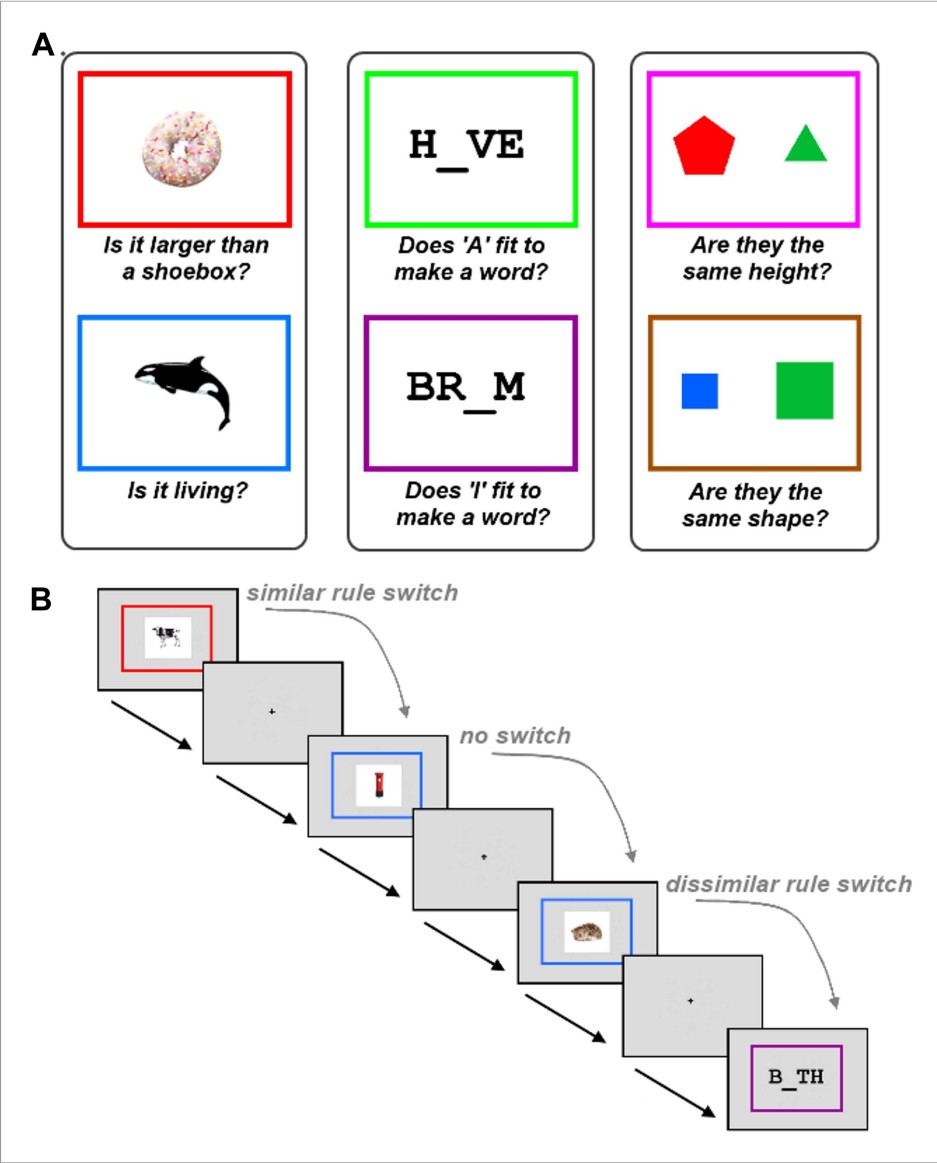

**Figure 1**. Task description. The experiment required participants to learn six tasks prior to scanning. (**A**) The six tasks were each associated with a different rule, as determined by the colour border. The tasks were split into three groups defined by stimulus category, with two possible tasks per stimulus type. (**B**) Experimental design. Within each run, trials using the six tasks occurred in random order. A no-switch trial occurred when participants had to apply the same task that was applied on the previous trial. A similar-task-switch trial occurred when participants had to apply the other task from the same category as the previous trial. A dissimilar-task-switch occurred when participants had to apply a task from a different category compared to the previous trial.

with switches between similar tasks (1746 ms; $t_{17} = 8.1$, p < 0.001). Switches between similar tasks also produced significantly longer RTs compared to no-switch trials ($t_{17} = 2.8$, p = 0.006).

## Task-switch related activity in the DMN

Preprocessing steps for fMRI data included realignment of the raw echo-planar images (EPI), slice-time correction, coregistration of the EPI images with the structural image, normalisation to the Montreal Neurological Institute (MNI) template brain, smoothing with an 8 mm full-width at half-maximum Gaussian kernel and filtering with a high-pass filter (see 'Materials and methods'). Univariate analysis of fMRI data was used to compare dissimilar-task-switch with no-switch trials through the

standard general linear model (GLM) approach. A regressor was constructed for each switch type with events modelled from stimulus onset until response and convolved with the haemodynamic response function. The resulting beta values for each switch type were compared and thresholded at p < 0.05, correcting for the false discovery rate. We identified widespread activation predominantly in regions of in the DMN (*Figure 2A*), with peaks found in bilateral HF, PHC, Rsp, PCC, AMPFC, and left pIPL (*Table 1*). It is worth noting that all of these regions fall within either the Core or MTL sub-networks. In comparison, no regions from the DMPFC sub-network showed significant activation at the whole-brain level. A contrast of similar-task-switch against no-switch trials revealed no significant activation across the whole brain.

To examine changes in activation from the perspective of the DMN sub-networks, we used individual DMN regions of interest (ROIs) previously defined (*Buckner et al., 2009*; *Andrews-Hanna et al., 2010*). The mean beta value was extracted from each ROI following each switch type. Planned, paired two-tailed t-tests revealed significant increase in activity during dissimilar-task-switch compared to no-switch in core (bilateral AMPFC, PCC) and MTL (Rsp, PHC) sub-networks, with a tendency to de-activation in the DMPFC sub-network (significant in right TPJ) (*Figure 2C*). Again, no ROIs revealed a significant difference between the similar-task-switch trials and no-switch trials. Two-way repeated measures ANOVAs were performed separately for each sub-network, with factors of ROI (Core: 4, MTL: 9, DMPFC: 7) and switch type (no-switch, dissimilar-task-switch). Main effects of task-switching were found for the Core ($F_{(1,17)}$ = 16.7, p = 0.001) and MTL ($F_{(1,17)}$ = 6.1, p = 0.03) sub-networks, showing increased activity for dissimilar switches. In contrast, the DMPFC sub-network

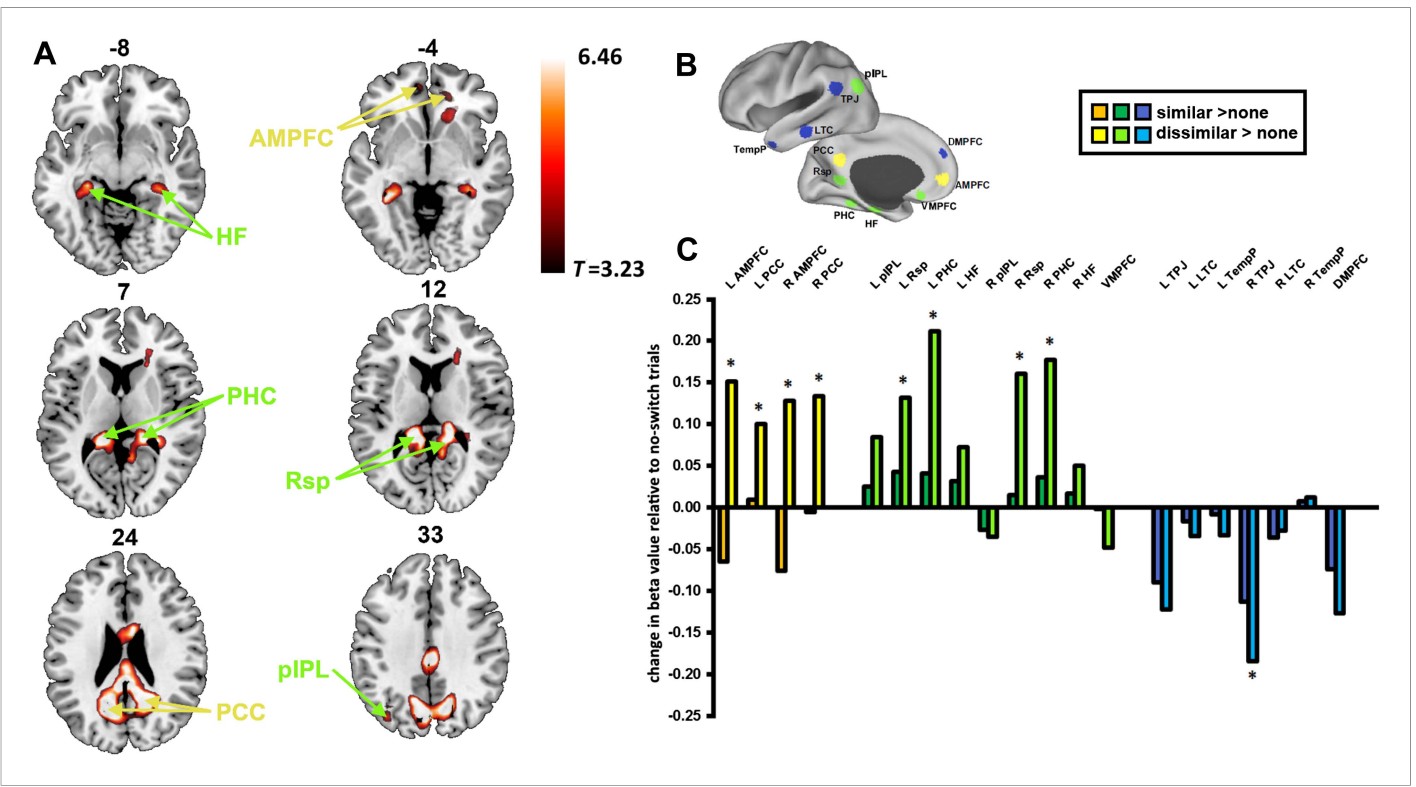

**Figure 2**. Activation of the default mode network (DMN) for dissimilar task switches. Region labels and regions of interest (ROIs) are color-coded according to the sub-network to which they belong: yellow for Core, green for medial temporal lobe (MTL), blue for dorsomedial prefrontal cortex (DMPFC). (**A**) Whole brain rendering in axial slices: the numbers above each slice indicate z-coordinate of that slice. The contrast of dissimilar-task-switch > no-switch (T = 3.23, p < 0.05, FDR corrected) shows activations in regions previously identified as the DMN. (**B**) Locations of DMN ROIs distinguished by Andrews-Hanna et al. (**C**) Change in activation of similar-task-switch (darker colours) and dissimilar-task-switch (lighter colours) relative to no-switch trials in the DMN ROIs. APMFC: anterior medial prefrontal cortex, PCC: posterior cingulate cortex, pIPL: posterior inferior parietal lobe, Rsp: retrosplenial cortex, PHC: parahippocampal cortex, HF: hippocampal formation, VMPFC: ventromedial prefrontal cortex, TPJ: temporoparietal junction, LTC: lateral temporal cortex, TempP: temporal pole, DMPFC: dorsomedial prefrontal cortex. * indicates p < 0.05.

**Table 1**. Peak coordinates of DMN regions that showed significantly greater activation for dissimilar-task-switch over no-switch

| ROI | hemisphere | x | y | z | t-statistic |
|---|---|---|---|---|---|
| HF | left | −30 | −36 | −6 | 3.64 |
| | right | 33 | −36 | −9 | 3.73 |
| PHC | left | −21 | −42 | 9 | 4.87 |
| | right | 30 | −39 | 6 | 6.81 |
| Rsp | left | −9 | −48 | 12 | 3.97 |
| | right | 9 | −51 | 12 | 3.89 |
| PCC | left | −12 | −54 | 24 | 5.12 |
| | right | 12 | −51 | 24 | 5.16 |
| AMPFC | left | −9 | 51 | −6 | 3.24 |
| | right | 9 | 48 | −3 | 3.72 |
| pIPL | left | −39 | −75 | 33 | 3.58 |

Coordinates are in MNI space. HF = hippocampal formation, PHC = parahippocampus, Rsp = retrosplenial cortex, PCC = posterior cingulate cortex, AMPFC = anterior medial prefrontal cortex, pIPL = posterior inferior parietal lobe.

showed a marginally significant de-activation ($F_{(1,17)}$ = 4.1, p = 0.06). Corresponding ANOVAs were performed to test for the difference between similar-task-switch and no-switch, but these revealed no main effect in any sub-network. To investigate differences at the sub-network level, beta values were averaged across the ROIs within each sub-network each of the three trial types, and a two-way repeated measures ANOVA (factors of sub-network and switch type) was performed on the mean beta values. This analysis revealed a main effect of sub-network ($F_{(2,34)}$ = 18.9, p < 0.001) and an interaction of switch type and sub-network ($F_{(4,68)}$ = 17.8, p < 0.001). These data therefore show a dissociation within the DMN: While the DMPFC sub-network displayed the characteristic pattern of reduced activity during executive control, switching between dissimilar tasks showed an opposite pattern of increased activity in Core and MTL sub-networks.

In an exploratory analysis, we looked at the univariate activation associated with dissimilar switches between specific categories compared to no-switch trials, in the three DMN sub-networks, for example, from a semantic task to a lexical task, compared with repetition of the lexical task. This was performed for all six between category switch types. *Figure 3* shows that all types of switch showed a relative increase in Core sub-network activation and decrease in DMFPC sub-network activation. The MTL sub-network shows increases for 4 of the 6 switch types and a marginal decrease when switching from the perceptual task. Especially for the Core and DMPFC sub-networks, these data suggest little variation in the pattern of results across different task types.

## Multivariate decoding demonstrates task representation across the DMN

For the multivariate analysis, the same preprocessing pipeline was followed with the omission of the smoothing step. We reasoned that if the DMN was involved in switching between tasks, then the differences between tasks might be represented within the network. To test this hypothesis, we performed a multivariate pattern analysis on the same ROIs. For each ROI, classifiers were trained to discriminate between the voxel-wise pattern of activation for each task pair (6 tasks, therefore 15 task pairs) and these classifiers were subsequently tested on independent data using leave-one-run-out cross-validation (see 'Materials and methods'). The matrices in *Figure 4A* show the classification accuracy (CA) for each task pair in each ROI, averaged across participants. The strongest decoding of task was found in bilateral HF, pIPL and the PCC, while bilateral TPJ, Rsp, AMPFC, and DMPFC on the midline showed weaker but still significant decoding.

Decoding of task between dissimilar task pairs is likely driven by many differences in task features. In contrast, differences between similar tasks will be predominantly driven by the internal representation of the specific decision rule. To quantify the extent to which rule and other features were driving the CA scores, CAs for ROIs within the Core, MTL and DMPFC sub-networks were averaged separately for similar task pairs and dissimilar task pairs. This analysis (*Figure 4B*) revealed significant decoding of dissimilar task pairs in all DMN sub-networks, and weaker but significant decoding of similar task pairs in the DMPFC sub-network.

Recently concern has arisen that differences in RT may be driving differences in CA (*Todd et al., 2013*; for contrary arguments see; *Woolgar et al., 2014*). We performed a regression analysis of CA against absolute difference in RT in each of the three sub-networks separately. First, we extracted the CA associated with each task pair in each ROI in each subject. We then calculated the mean CA across the component ROIs of the Core, MTL, and DMPFC sub-networks in each individual, producing

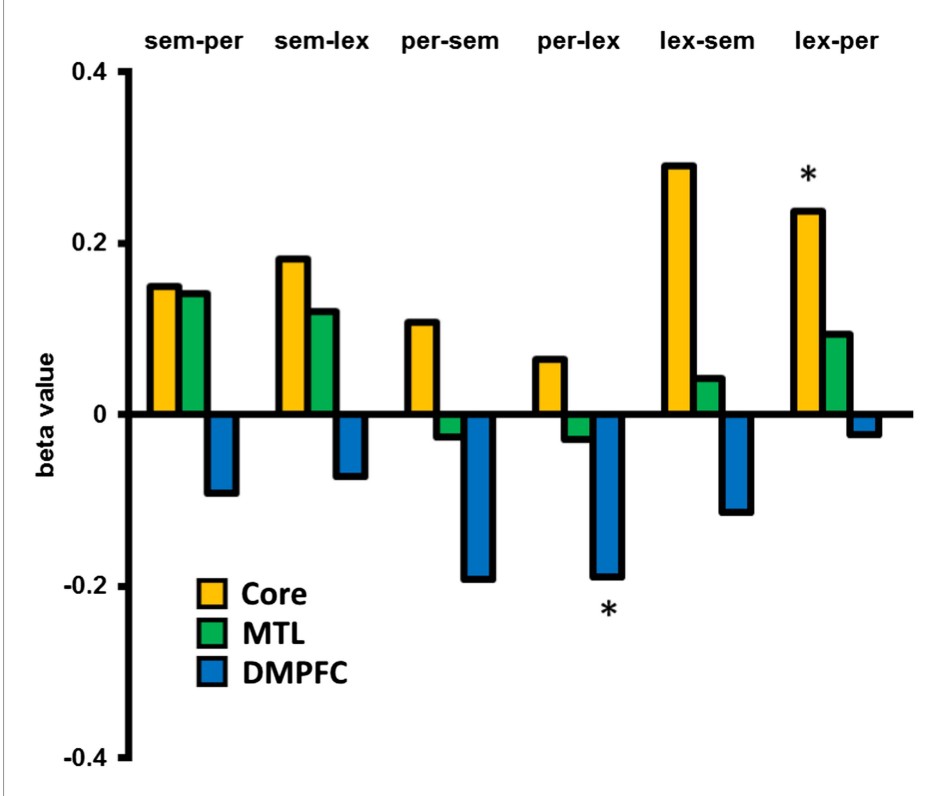

**Figure 3**. Activation associated with each between category switch. An exploratory analysis looking at the activation/deactivation associated with switching between each of the three task categories. Core and MTL sub-networks predominantly show increased activation following a dissimilar task switch across switch types, whereas DMPFC shows a relative decrease in activation. Abbreviations: sem = semantic category, per = perceptual, lex = lexical. * denotes $p < 0.05$ from a paired, two-tailed t-test.

a 3-dimensional matrix of CA values for 3 sub-networks × 18 subjects × 15 task pairs. A similar matrix was produced for absolute RT differences. We segregated the dissimilar–task pairs and similar–task pairs and conducted a Spearman's correlational analysis of RT against CA for each task-pair type within each sub-network. The results showed strong discrimination of dissimilar task pairs, and weak discrimination of similar task pairs, irrespective of RT difference (*Figure 5A*).

A second analysis considered data from each subject separately, with a separate regression analysis for similar and dissimilar task pairs in each ROI. A general linear model was constructed with a regressor for the absolute RT difference for either the similar-task-pairs or dissimilar-task-pairs, which was fit to the corresponding CA data. This produced a beta estimate for similar and dissimilar-task-pairs in each ROI, in each subject. Beta estimates were subsequently averaged across the component ROIs of each sub-network. *Figure 5B* shows the mean beta estimate for the similar and dissimilar task pairs in each subject for the Core, MTL, and DMPFC sub-networks. Within each graph, data from the 18 subjects are sorted in ascending order. Overall, the graphs suggest that RT does not systematically predict CA across participants, especially for the dissimilar task pairs (bottom row) for which CA was highest.

Together these analyses demonstrate that simple RT differences were not strongly driving the classification accuracies.

## Discussion

In previous work, DMN activity has been associated with a variety of complex, often self-referential mental processes, such as retrieving past events from one's life, imagining possible future events, or considering the beliefs of oneself and others (*Buckner and Carroll, 2007*). Here, we argue that a simple variable may relate these complex processes—the degree of change in cognitive context.

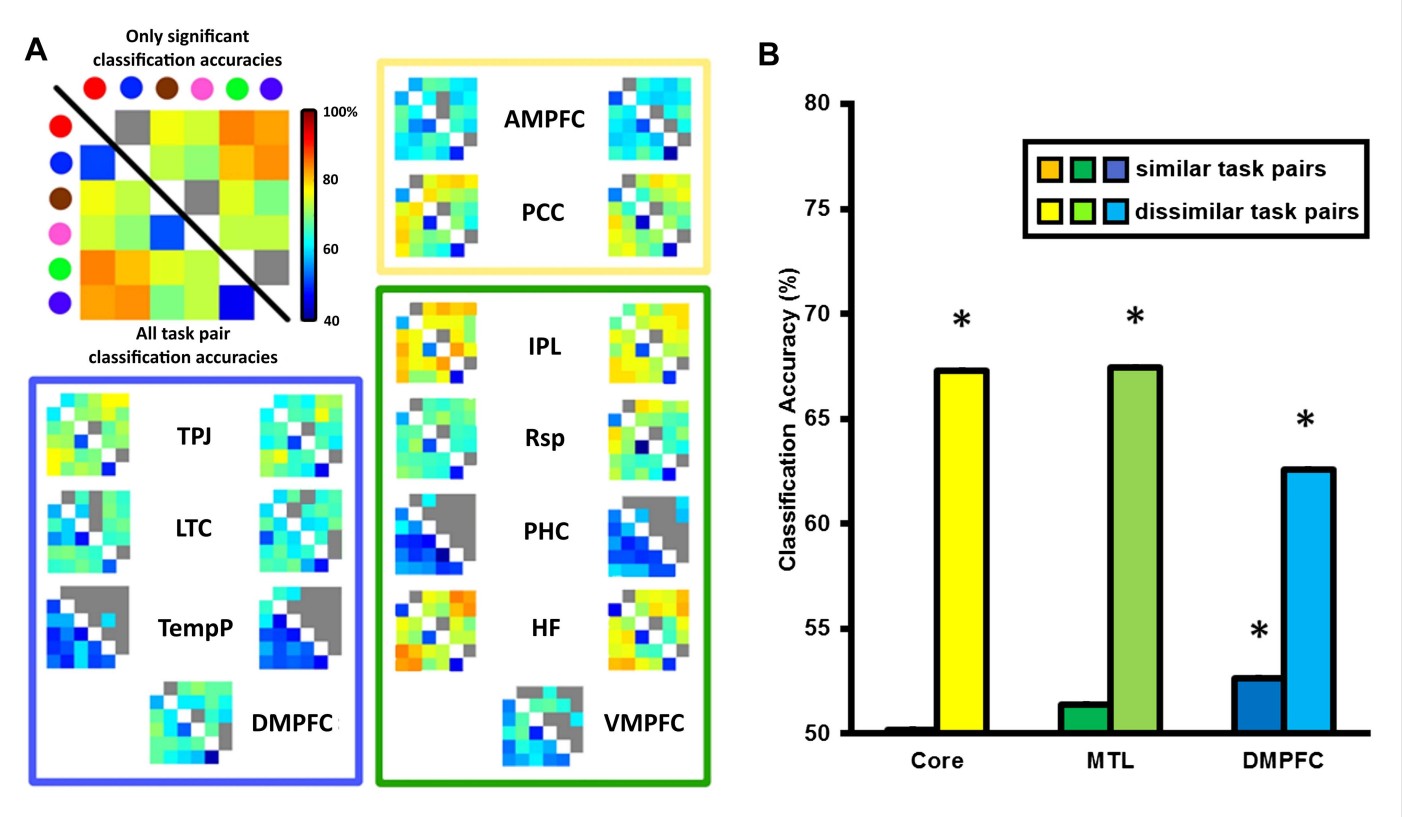

**Figure 4**. Classification accuracy (CA) within the DMN sub-network ROIs. (**A**) Classification accuracies between different task pairs in all DMN ROIs. Large correlation matrix used as example is the same as left HF. The colour of the circle in the key matches with the colour borders used to indicate each task in *Figure 1A*: Red for 'bigger than a shoebox?', blue for 'living?', brown for 'same shape?', pink for 'same height?', green for 'does adding A make a word?', purple for 'does adding I make a word'? Matrices show the classification accuracy of decoding each task pair; values below the diagonal show classification accuracy for all task pairs, while non-grey values above the diagonal show only decoding that survived the threshold for statistical significance. The colour borders indicate the sub-network that the ROIs belong to: core (yellow), MTL (green), and DMPFC (blue). ROIs on the left side of each box are from the left hemisphere, those on the right are from the right hemisphere. (**B**) All three sub-networks demonstrated above-chance classification accuracy when decoding dissimilar tasks, while only the DMPFC sub-network demonstrated significant decoding of similar task pairs. Error bars indicate standard error. * indicates $p < 0.05$.

To address this hypothesis we modified a typical task switching study to include both small shifts—similar to those of many previous studies—and much larger shifts. In contrast to the common idea of the DMN as a task-negative system, the activity of which progressively decreases with increasing task difficulty, our results show the opposite for a large change of cognitive context, in particular for Core and MTL sub-networks.

As cognitive context is changed, elements of the old context must be suppressed and elements of the new context must be retrieved or activated, producing a reconfiguration appropriate to the new circumstances. Our data suggest DMN activation only when the change is sufficiently large, perhaps analogous to many of the shifts taking place in everyday cognition. It is uncertain whether large and small shifts differ qualitatively or only quantitatively. In our task, for example, a shift between rules within the same category may have been executed without reference to the link between frame color and categorization rule; to perform such a switch, it was necessary only to see that frame color had changed and to retrieve the other rule relevant to the current set of stimuli. In contrast, a shift between categories likely required reference back to a broader set of task rules, including the remembered list of color–rule combinations. More broadly, one possibility is that the DMN is involved in relaxing a current attentional focus, allowing new cognitive contents to arise. When cognitive operations are largely similar across successive trials, the DMN may be suppressed, but as increasingly more of the current focus must be dissolved, suppression may shift to activation.

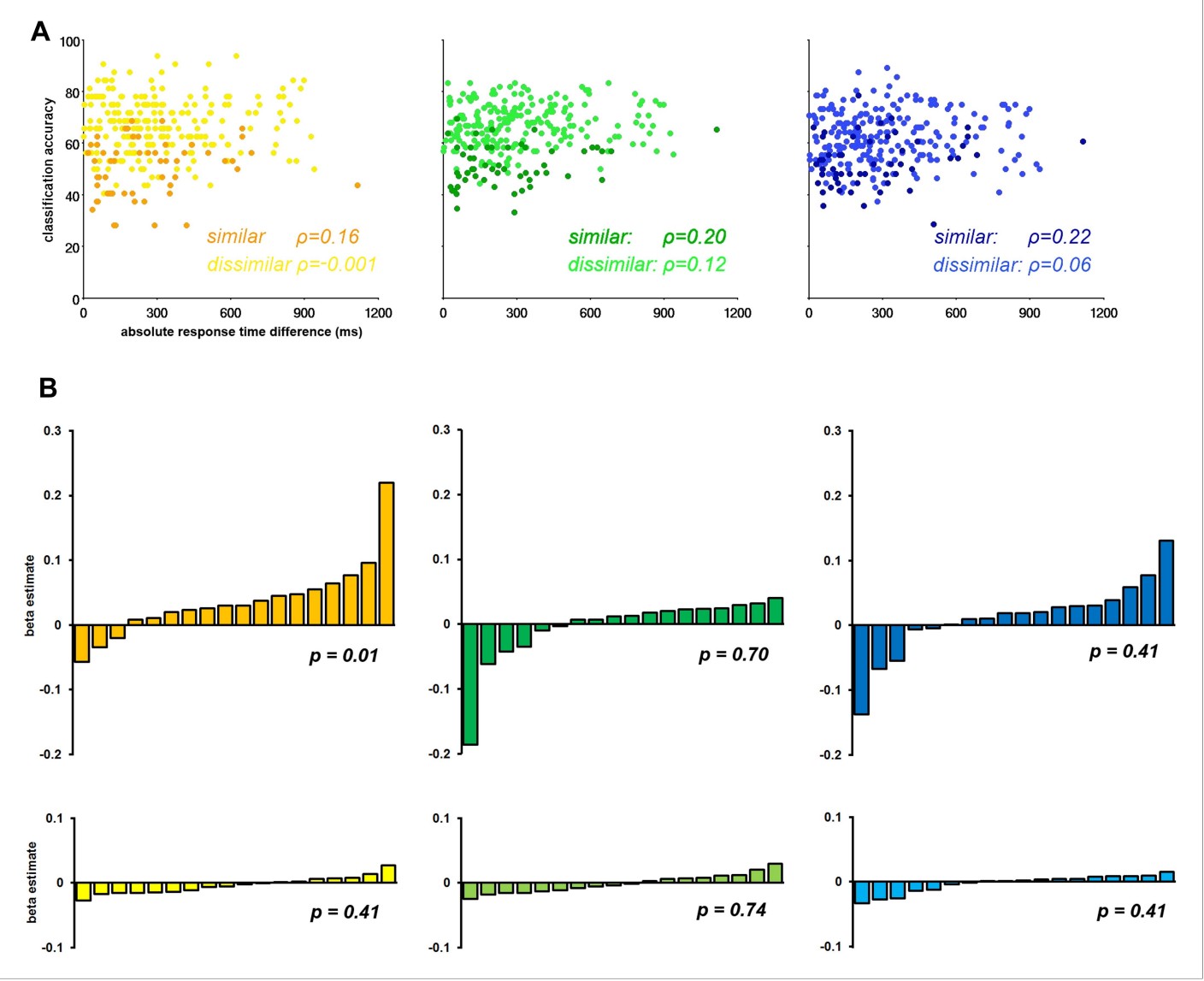

**Figure 5**. The influence of response time (RT) on classification accuracy. (**A**) Correlation between classification accuracy and RT difference in the three DMN sub-networks. Each point represents data for a single task pair in a single subject, with mean CA across ROIs of the named sub-network plotted against absolute RT difference. The darker shades in each graph are taken from the similar task pairs, while the lighter shades are taken from the dissimilar task pairs. (**B**) Beta estimates for the association of CA and RT in each subject for similar and dissimilar task pairs of the three DMN sub-networks. In each graph subjects' beta estimates are sorted is in ascending order. Top row, a–c, displays beta estimates for similar task pairs in the Core, MTL and DMPFC, respectively. Bottom row, d–f, shows beta estimates for dissimilar task pairs in the Core, MTL and DMPFC, respectively. The p-value from a 2-tailed, 1-way t-test of each graph's beta values is shown.

Thus, it is the magnitude of switch involved that distinguishes this study from much of the previous work in the literature. The point is illustrated by a detailed consideration of the task-switching review by *Kim et al. (2011)*, as described in the introduction. *Kim et al. (2011)* identified three major types of task-switch found in the literature: perceptual switch, response switch, and contextual switch, the latter of which is the most comparable to the current work. Of the 20 tasks that they identified as involving a contextual switch, 17 used fixed small stimulus sets, with the switches (like our within-category switches) concerning just the relevant stimulus feature such as colour or shape. Two studies used a task that required participants to make one of two possible binary decisions on serially presented letter strings. The remaining task was essentially our semantic task, using text instead of images. Unlike these studies, we propose that it is our use of much more substantial switches, requiring larger revisions of cognitive context and operations, that leads to the novel finding of DMN activation.

Much previous work links activity in the DMN to conscious recollection, commonly defined by the ability to link a remembered item to the surrounding context of a specific event. A role in memory for individual events fits well with the proposal that the DMN binds together the components of a complex cognitive context. Beyond traditional studies of episodic and autobiographical memory, our results show the importance of DMN context processing in cognitive control. In contrast to the common finding of deactivation linked to executive control, our findings show increased activity when the change of context is sufficiently large. Establishing a new, complex context, we suggest, may be common to recollection of specific previous events and to major revisions of complex task rules.

An intriguing aspect of the present results is the difference between the regions that show switch-related activity and those that encoded task-related information. The regions that demonstrate switch-related activity show relatively strong correspondence to the pattern of DMN fractionation presented by Andrews-Hanna et al. In contrast, the DMN regions that show strong encoding extend across all three sub-networks, suggesting only partial functional segregation. Encoding analysis is likely more sensitive than univariate activity analysis, reflecting both overall activity within a broad region, and the exact pattern of activity within that region (*Davis et al., 2014*). Though DMPFC showed no univariate signal linked to switching task category, our results suggest some involvement in task representation.

Our results show very different levels of CA for discrimination of similar vs dissimilar tasks. For the comparison of similar tasks, other than the different colour of border surrounding the task item, the only difference is the internal representation of task rule. In contrast, comparing dissimilar tasks involves a myriad of differences such as visual properties of the stimuli, cognitive domain (semantic knowledge, lexicon, visual discrimination) as well as task rule. Correspondingly, much stronger decoding is seen across the DMN between dissimilar tasks. This is unsurprising for a network of regions hypothesised to link components of a broad cognitive context.

Many studies suggest sustained DMN activity in rest compared to active task performance. It is unclear how such sustained activity relates to the switch-related activity we have shown here. On the one hand, it is plausible that, when a participant lies in a scanner at rest, there are periodic large shifts in the content of cognition, and in part, 'sustained' DMN activity may reflect averaging across shifts occurring at variable, unknown times. Indeed, traditional studies of resting state functional connectivity depend on temporal variation in network activity, as expected for a signal in part linked to transient cognitive events. That said, if a core aspect of DMN function is relaxing an attentional focus, sustained enhancement is plausible during a period of relatively unfocused cognitive activity.

An unexpected aspect of our results is the lack of switch-associated activity when changing between similar tasks, which does show a robust behavioral cost compared to no-switch trials. In previous studies, apparently similar cases of task switching have been associated with widespread recruitment of a fronto-parietal, executive control network (*Sohn et al., 2000*; *Braver et al., 2003*; *Monsell, 2003*; *Yeung et al., 2006*; *Kim et al., 2011*), and it is unclear why no similar activity was seen in our data. One contributing factor may be our explicit modelling of RT differences between conditions, convolving the canonical haemodynamic response with the duration of each trial from stimulus presentation until response. As this procedure is designed to correct for activation differences due simply to longer RT on switch trials, it may reduce or remove differences seen in studies that do not adopt such a correction. Our results also raise the possibility, however, that traditional task-switching results may not generalise to the more complex conditions of our experiment. Future work will be needed to resolve this discrepancy.

## Conclusion

In conclusion, we propose that the DMN may be recruited whenever large changes of cognitive context are required. This may apply in complex cases of self-referential processing, mind wandering etc, but also in relatively simple acts of cognitive or executive control. The DMN, widely seen as a 'task-negative' network, may respond positively to any task which demands a switch from one broad context to another.

## Materials and methods

### Participants

18 right-handed participants (10 females) aged between 18 and 40 were recruited from the Medical Research Council Cognition and Brain Sciences Unit subject panel. Of 21 original subjects scanned,

three had to be removed for excessive head-movements (over 10 mm translation and/or 6° rotation). No participant had a history of neurological or psychiatric illness. Participants were reimbursed for their time. Ethics approval was given by the Cambridge Psychology Research Ethics Committee.

## Task description

All three tasks were created using the Psychophysics Toolbox for MATLAB (*Brainard, 1997*). Within the scanner, the stimulus display was projected onto a mirror mounted to a 32-channel head-coil.

Participants were required to learn six different tasks (*Figure 1A*). Each task was associated with a different rule, with the appropriate rule determined by the colour border in which the task stimulus appeared. The six tasks rules are shown in *Figure 1A*. Prior to scanning, participants practised the task until they had completed at least 20 trials with an accuracy exceeding 80%. Importantly, the six tasks were grouped into three categories of two tasks each, where the stimuli within a category could be relevant to either rule within that category, but not to rules of other categories. Furthermore, all categories included trials which required a positive answer for both rules, for one rule but not the other, or for neither rule; therefore subjects needed to remember and apply the correct rule on all trials. All questions were framed in a true/false format, so that arbitrary response mappings for each rule did not have to be learned in addition to the rules themselves.

Each trial began with the simultaneous appearance of the colour border (visual angle = 7.9°) and the task stimulus (*Figure 1B*). Participants were requested to respond as quickly as possible with a true or false answer (right thumb button press = true, left thumb button press = false). The border and stimulus remained on screen until the subject had made their response. A low tone was played to participants if they made an incorrect response. There was a jittered interval between the response to one trial and the onset of the next. Interval jittering followed an exponential distribution between 1 s and 11 s, with a mean of 4.1 s.

Participants learned the tasks prior to scanning. An event-related design was adopted, with 73 trials per run. Each run had at least 12 trials of each of the six task types. Task switch type was also balanced within a run: 24 no-switch trials, 24 similar-task-switches, and 24 dissimilar-task-switches. Post scanning, when questioned, no participants reported having any sense of what task to expect on a given trial.

## fMRI acquisition

Scans were acquired with a 3T Siemens Trim Trio scanner. 32 3-mm slices (0.75 mm interslice gap) in axial orientation gave an in-plane resolution of 3 × 3 mm and were acquired using a TR of 2 s. T2*-weighted EPI capturing blood oxygen level dependent contrast was employed with a flip angle of 78°. For both experiments, the first eight images were discarded to avoid T1 equilibration effects.

## Univariate analysis

For the univariate analysis, images were preprocessed and analysed with SPM5 (Wellcome Department of Cognitive Neurology). In the first preprocessing step, data were checked for obvious artefacts, and all images were realigned to the first image. Next we performed slice time correction and coregistration of the structural with the functional EPI images. Finally, data were normalized to the standard MNI template, smoothed with an 8 mm full-width at half-maximum Gaussian kernel and subjected to a high-pass filter with cut-off at 128 s.

Fixed-effects analyses were performed on each individual's data using a general linear model. In the first univariate analysis investigating switching related activity three regressor functions were used (no-switch trials, similar-task-switch trials, and dissimilar-task-switch trials). Each regressor was modelled as a rectangular function from the onset of each stimulus to the moment of response and convolved with the canonical hemodynamic response function. Beta weight images were contrasted for the conditions dissimilar-task-switch > no-switch and similar-task-switch > no-switch. Contrasts were further examined by random-effects analysis. Activation maps (threshold 0.05, FDR-corrected) were visualised using MRIcroGL (*Rorden et al., 2007*).

For ROI analysis, mean contrast values within each ROI were extracted for each subject, using the MarsBaR SPM toolbox (*Brett et al., 2002*). ROIs were spherical, with 8 mm radius, based around peak coordinates (*Figure 2B*) taken from *Andrews-Hanna et al. (2010)*.

For the exploratory analysis into the activation associated with switches between specific categories a separate GLM was constructed. 36 regressors were used (one regressor for each possible switch type) and modelled as before. The resulting beta estimates were then processed using the same ROI analysis method as before with the same ROIs.

## Multivoxel pattern analysis

Multivoxel pattern analysis was performed using the Decoding Toolbox (*Christophel et al., 2012*; *Görgen et al., 2012*). Preprocessing of the data was the same as for the univariate whole-brain analysis, except for the omission of the smoothing step. Again, a fixed effects analysis was performed on each participant's data using a general linear model. For this GLM, each task was modelled as a separate regressor, constructed as a rectangular function from the onset of each stimulus to the moment of response and convolved with the canonical hemodynamic response function. The same ROIs as previously (*Figure 2B*) were used.

Prior to pattern analysis, beta values were Z-scored across tasks within each voxel of the ROI. This step was intended to reduce any impact of task differences in overall ROI activity. Pattern discrimination between tasks was then estimated using pairwise classification, that is, only 1 of the 15 possible task pairs was decoded at a time. A support vector machine (LIBSVM) (*Fan et al., 2005*) was used to train and classify data from three of the four runs, with the remaining run used to test the classifier. Test and training runs were always kept separate and each run was used to test the classifier once, that is, fourfold cross-validation. The CA for a given ROI was averaged across test-train splits, yielding a single CA for each ROI, in each individual, for each task pair.

## Acknowledgements

This work was funded by the Medical Research Council (UK) intramural program MC_US_A060_0001.

## Additional information

### Funding

| Funder | Grant reference | Author |
| --- | --- | --- |
| Medical Research Council (MRC) | MC_US_A060_0001 | John Duncan |

The funder had no role in study design, data collection and interpretation, or the decision to submit the work for publication.

### Author contributions

BMC, Conception and design, Acquisition of data, Analysis and interpretation of data, Drafting or revising the article; DJM, Analysis and interpretation of data, Drafting or revising the article; JD, Conception and design, Analysis and interpretation of data, Drafting or revising the article

### Author ORCIDs

Daniel J Mitchell, http://orcid.org/0000-0001-8729-3886

### Ethics

Human subjects: Informed consent, and consent to publish, was obtained through the University of Cambridge ethics committee: CPREC (Cambridge Psychology Research Ethics) 2010.16.

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
