## [Decision Letter]

Thank you for sending your work entitled “Recruitment of the default mode network during a demanding act of executive control” for consideration at *eLife*. Your article has been favorably evaluated by Eve Marder (Senior editor), a Reviewing editor, and two reviewers.

The Reviewing editor and the reviewers discussed their comments before we reached this decision, and the Reviewing editor has assembled the following comments to help you prepare a revised submission.

This study provides evidence that parts of the default mode network (DMN) are activated when subjects switch between dissimilar tasks. It is a well-done study that has potentially interesting implications for our understanding of the cognitive functions of the default mode network. Because the function of the DMN is controversial, any demonstration that a novel paradigm produces robust DMN activations above a resting baseline is of broad interest and a potentially important advance. However, the reviewers share conceptual and other concerns about the paper as it stands.

1) A major concern is the disparity between your results and previous task-switching studies. In the 36-study meta-analysis you cite (15), activations are generally observed in dorsolateral prefrontal cortex and posterior parietal cortex, among other regions, rather than the DMN. In contrast, your study reports activity in some parts of the DMN but no activity in regions previously associated with task switching. You propose that the DMN is activated when a change of context is ‘sufficiently large’, but it is unclear whether the change of context is much larger (or how much larger) than the change in all the other studies that have been done.

The paper would be stronger if you could demonstrate a modification of the current experimental paradigm that controls whether activations are observed in the DMN or in the regions previously reported in the literature. For example, are DMN activations observed for large changes in task context, while the standard pattern can be confirmed for a paradigm involving small changes? We recognize this would likely entail additional experimentation and analysis. Hence, it is made as a suggestion rather than a requirement for resubmission and re-review.

If you elect not to carry out additional experiments, we strongly encourage you to include a more systematic comparison of your paradigm in relation to earlier task-switching paradigms. Currently, this section of the Discussion is unduly cursory. You indicate that you do not know why similar task switches did not activate standard task switching regions, but do not consider the converse issue, i.e. why previous task-switching studies did not activate the DMN. Please include a literature review that supports the implication that previous studies did not involve a “large switch of cognitive context” or “a switch from one broad context to another”.

2) Your interpretation is somewhat in conflict with the results. The primary result is that DMN activity increases when one switches between the processing of two different stimulus classes (i.e., “dissimilar” switches). The fact that there is no effect of changing the question that is being asked (i.e., “similar” switch trials) about the same stimulus class seems to undercut the notion that activity in these regions are sensitive to changes in “cognitive context” since one would expect context to change between the different questions about the same stimulus type nearly as much as between different questions about a different stimulus class. The labeling of the two conditions as “similar” and “dissimilar” also contributes to the confusion by obfuscating the fact that it's really a change in the question vs a change in the stimulus type (in addition to the question).

3) The multivariate analysis is a useful confirmation of the univariate results, but not much more. There are some differences between univariate and multivariate analyses, but unfortunately these differences are basically uninterpretable, since they can arise either from differences in the nature of neural representation or from differences in the nature of variability in univariate signals in the data (e.g. between vs within subject variability; see http://www.ncbi.nlm.nih.gov/pubmed/24768930). Accordingly, we recommend that you remove or truncate the speculation on the implications of the MVPA analyses (e.g. the comparison of HF vs PHC), which is difficult to interpret because of the large differences between the ‘dissimilar’ task conditions. Many other regions, including ‘standard’ task-switching regions, might also successfully discriminate the dissimilar task conditions in MVPA analyses.

4) It seems possible that the observed effects could have been driven by changes between specific categories (e.g., changing from words to objects or vice versa) but you do not report any analyses of the different kinds of switches. This seems like it could provide important leverage on understanding the mechanisms at play.

5) You suggest that task switching accounts for the other circumstances in which DMN activations have been reported. However, it is unclear whether DMN activations are necessarily transient, as a task switching explanation would seem to imply. Please comment on this issue and also why some previous accounts of DMN function, such as those involving mind-wandering, introspection, or social cognition, seem only loosely related to task switching.

---

## [Author Response]

*This study provides evidence that parts of the default mode network (DMN) are activated when subjects switch between dissimilar tasks. It is a well-done study that has potentially interesting implications for our understanding of the cognitive functions of the default mode network. Because the function of the DMN is controversial, any demonstration that a novel paradigm produces robust DMN activations above a resting baseline is of broad interest and a potentially important advance. However, the reviewers share conceptual and other concerns about the paper as it stands*.

*1) A major concern is the disparity between your results and previous task-switching studies. In the 36-study meta-analysis you cite (*[15]*), activations are generally observed in dorsolateral prefrontal cortex and posterior parietal cortex, among other regions, rather than the DMN. In contrast, your study reports activity in some parts of the DMN but no activity in regions previously associated with task switching. You propose that the DMN is activated when a change of context is ‘sufficiently large’, but it is unclear whether the change of context is much larger (or how much larger) than the change in all the other studies that have been done*.

*The paper would be stronger if you could demonstrate a modification of the current experimental paradigm that controls whether activations are observed in the DMN or in the regions previously reported in the literature. For example, are DMN activations observed for large changes in task context, while the standard pattern* can *be confirmed for a paradigm involving small changes? We recognize this would likely entail additional experimentation and analysis. Hence, it is made as a suggestion rather than a requirement for resubmission and re-review*.

*If you elect not to carry out additional experiments, we strongly encourage you to include a more systematic comparison of your paradigm in relation to earlier task-switching paradigms. Currently, this section of the Discussion is unduly cursory. You indicate that you do not know why similar task switches did not activate standard task switching regions, but do not consider the converse issue, i.e. why previous task-switching studies did not activate the DMN. Please include a literature review that supports the implication that previous studies did not involve a “large switch of cognitive context” or “a switch from one broad context to another”*.

We agree that it is important to put our findings into the context of previous neuroimaging studies of task switching, and to this end we are grateful for the suggestion that we make use of the Kim et al. review. Based on this review, we have added new material to both Introduction and Discussion that we hope clarifies the relationship of ours study to previous work, as well as the theoretical interpretation we are proposing.

The first and more important question concerns our positive finding of DMN activity linked to a large shift of task. Our intention in this study was to introduce more substantial switches than previously used—more in line with many switches in everyday cognition—, and the Kim et al. review shows that we were largely successful in this. Of the 20 studies they review, 17 used fixed small stimulus sets, with the switches (like our within-category switches) concerning just the relevant stimulus feature such as colour or shape. Two studies used a task that required participants to make one of two possible binary decisions on serially presented letter strings. The remaining task was essentially our semantic task, using text instead of images. As we now discuss, we propose that it is our use of much more substantial switches, requiring larger revisions of cognitive context and operations, that leads to the novel finding of DMN activation. This conclusion is directly supported by our explicit comparison of smaller with larger switches.

A second question is why, in our study, we do not observe extensive activation of the classical frontoparietal executive network, either for large or for small switches, activation commonly reported in previous studies. As we discuss the explanation for this discrepancy is uncertain, though we do discuss one possible methodological contributor. This question, in any case, is tangential to the main point of the current study, and we feel that here we can leave it unresolved.

*2) Your interpretation is somewhat in conflict with the results. The primary result is that DMN activity increases when one switches between the processing of two different stimulus classes (i.e., “dissimilar” switches). The fact that there is no effect of changing the question that is being asked (i.e., “similar” switch trials) about the same stimulus class seems to undercut the notion that activity in these regions are sensitive to changes in “cognitive context” since one would expect context to change between the different questions about the same stimulus type nearly as much as between different questions about a different stimulus class. The labeling of the two conditions as “similar” and “dissimilar” also contributes to the confusion by obfuscating the fact that it's really a change in the question versus a change in the stimulus type (in addition to the question)*.

A second question raised by the reviewers concerns theoretical distinctions between large and small switches. Again, we hope that our points are now clarified by the Discussion of our study in the context of the previous literature, and in particular how our “dissimilar” switches relate both to our own “similar” switches and to previous work (Introduction and Discussion sections). We now discuss the way similar and dissimilar switches may differ in our study, and more generally, the factors that may promote the transition from DMN suppression to activation.

*3) The multivariate analysis is a useful confirmation of the univariate results, but not much more. There are some differences between univariate and multivariate analyses, but unfortunately these differences are basically uninterpretable, since they* can *arise either from differences in the nature of neural representation or from differences in the nature of variability in univariate signals in the data (e.g. between vs. within subject variability; see*
*http://www.ncbi.nlm.nih.gov/pubmed/24768930**). Accordingly, we recommend that you remove or truncate the speculation on the implications of the MVPA analyses (e.g. the comparison of HF vs PHC), which is difficult to interpret because of the large differences between the ‘dissimilar’ task conditions. Many other regions, including ‘standard’ task-switching regions, might also successfully discriminate the dissimilar task conditions in MVPA analyses*.

We have now removed the paragraph that speculates on the difference between HF and PHC, which we previously argued were suggested by the difference between the univariate and multivariate analyses. We have also included a reference to Davis et al.’s work comparing the two types of analysis and clarified our interpretation of these differences in line with this study (Discussion section).

*4) It seems possible that the observed effects could have been driven by changes between specific categories (e.g., changing from words to objects or vice versa) but you do not report any analyses of the different kinds of switches. This seems like it could provide important leverage on understanding the mechanisms at play*.

In response to this question, we add more information on switches between the different pairs of stimulus categories (subsection “Task-switch related activity in the DMN”). Although not statistically significant due to low power, the trends are in the expected direction. Strengthening our conclusions, these analyses confirm that DMN activation is associated with all possible pairs of between-category switches.

*5) You suggest that task switching accounts for the other circumstances in which DMN activations have been reported. However, it is unclear whether DMN activations are necessarily transient, as a task switching explanation would seem to imply. Please comment on this issue and also why some previous accounts of DMN function, such as those involving mind-wandering, introspection, or social cognition, seem only loosely related to task switching*.

We have offered some discussion on this issue in the penultimate paragraph of the Discussion, relating sustained DMN activity during rest both to occasional transient events and to sustained maintenance of an open cognitive focus.